# Recognizing Pediatric Tuberous Sclerosis Complex Based on Multi-Contrast MRI and Deep Weighted Fusion Network

**DOI:** 10.3390/bioengineering10070870

**Published:** 2023-07-22

**Authors:** Dian Jiang, Jianxiang Liao, Cailei Zhao, Xia Zhao, Rongbo Lin, Jun Yang, Zhi-Cheng Li, Yihang Zhou, Yanjie Zhu, Dong Liang, Zhanqi Hu, Haifeng Wang

**Affiliations:** 1Research Centre for Medical AI, Shenzhen Institutes of Advanced Technology, Chinese Academy of Sciences, Shenzhen 518000, China; dian.jiang@siat.ac.cn (D.J.); jun.yang@siat.ac.cn (J.Y.); zc.li@siat.ac.cn (Z.-C.L.); yihang.zhou@outlook.com (Y.Z.); dong.liang@siat.ac.cn (D.L.); 2University of Chinese Academy of Sciences, Beijing 100049, China; yj.zhu@siat.ac.cn; 3Department of Neurology, Shenzhen Children’s Hospital, Shenzhen 518000, China; liaojianxiang@vip.sina.com (J.L.); 837623191@qq.com (X.Z.); 4Department of Radiology, Shenzhen Children’s Hospital, Shenzhen 518000, China; zhaocailei197866@163.com; 5Department of Emergency, Shenzhen Children’s Hospital, Shenzhen 518000, China; 397126778@qq.com; 6Research Department, Hong Kong Sanatorium & Hospital, Hong Kong 999077, China; 7Paul C. Lauterbur Research Center for Biomedical Imaging, Shenzhen Institutes of Advanced Technology, Chinese Academy of Sciences, Shenzhen 518000, China

**Keywords:** tuberous sclerosis complex, children, convolutional neural network, multi-contrast MRI, rare neurodevelopmental disorder

## Abstract

Multi-contrast magnetic resonance imaging (MRI) is wildly applied to identify tuberous sclerosis complex (TSC) children in a clinic. In this work, a deep convolutional neural network with multi-contrast MRI is proposed to diagnose pediatric TSC. Firstly, by combining T2W and FLAIR images, a new synthesis modality named FLAIR_3_ was created to enhance the contrast between TSC lesions and normal brain tissues. After that, a deep weighted fusion network (DWF-net) using a late fusion strategy is proposed to diagnose TSC children. In experiments, a total of 680 children were enrolled, including 331 healthy children and 349 TSC children. The experimental results indicate that FLAIR_3_ successfully enhances the visibility of TSC lesions and improves the classification performance. Additionally, the proposed DWF-net delivers a superior classification performance compared to previous methods, achieving an AUC of 0.998 and an accuracy of 0.985. The proposed method has the potential to be a reliable computer-aided diagnostic tool for assisting radiologists in diagnosing TSC children.

## 1. Introduction

Tuberous sclerosis complex (TSC) is a rare neurodevelopmental disorder caused by mutations in the TSC1 and TSC2 genes [1,2]. It is characterized by angiofibromas of the face, epilepsy, an intellectual disability, and hamartomas in multiple organs including the heart, kidneys, brain, and lungs [3,4,5]. The majority of pediatric TSC patients experience their initial seizure in the first year of life [6,7,8], which has a severe impact on the lives of TSC children [9,10]. Therefore, it is urgent and valuable to develop valid and robust classification models for TSC children in a clinic.

Neurological symptoms are prevalent in nearly all children with TSC, and multi-contrast magnetic resonance imaging (MRI) is frequently employed for a clinical diagnosis [11]. To date, T2-weighted imaging (T2W) and fluid-attenuated inversion recovery (FLAIR) have been commonly utilized in a pediatric TSC diagnosis, allowing for the identification of lesions and facilitating high lesion-to-brain contrast visualization. But, the cerebrospinal fluid (CSF) signal is strong in T2W, which severely interferes with the visualization of periventricular TSC lesions. FLAIR imaging can suppress cerebrospinal fluid and sufficiently show the lesion–brain contrast clearly, and FLAIR also reduces the signal-to-noise ratio while pressing CSF [12]. Currently, it is not possible for a single MRI sequence to produce all the required tissue contrasts in a single contrast image due to the trade-offs that need to be made when choosing MRI pulse sequence parameters [13]. In recent studies, it has been demonstrated that a synthesized contrast that blends T2W and FLAIR imaging can augment the contrast of multiple sclerosis (MS) lesions, leading to an improved diagnostic efficacy [12,13]. However, to the best of our knowledge, there are not studies on applying a synthesis contrast combining T2W and FLAIR for diagnosing pediatric TSC so far.

Otherwise, deep learning has been studied as an advanced artificial intelligence technology that can automatically learn from medical image data and extract a large number of features [14]. Previously, deep learning models and multi-contrast MRIs have been successfully used for automatically detecting strokes [15] and classifying brain tissues [16]. Until now, convolutional neural networks (CNNs) have been applied to assist in tuber segmentation in TSC patients [17]. Sanchez et al. [18] used two types of contrast MRI, T2W and FLAIR, for the detection task of TSC tubers and achieved the receiver operating characteristic curve that can have an area under the curve (AUC) of 0.99. However, their approach employed a 2D network and solely relied on handpicked MRI slices with evident tubers as input to the network. This method failed to account for the spatial attributes of MRI and neglected the fact that not all TSC patients exhibit visible lesions. Additionally, their datasets were limited to merely 114 TSC patients and 114 controls. Alternatively, recent research suggests that 3D CNNs excel at capturing the spatial characteristics of MRI and effectively capitalize on the interplay between voxels. Consequently, they have been reported to yield superior results in predicting chronological age [19].

To further raise the performance of identifying TSC children in a clinic, a novel deep learning method, named the deep weighted fusion network (DWF-net), was proposed to effectively diagnose pediatric TSC lesions with multi-contrast MRIs. The proposed method has a synthesis contrast, named FLAIR_3_, from the combination of T2W and FLAIR that can maximize the lesion–brain contrast of pediatric TSC lesions. Moreover, the proposed method has a 3D CNN strategy of the weighted late fusion model combined with multi-contrast MRI to automatically diagnose pediatric TSC. The experimental dataset has a total of 680 children, including 331 healthy and 349 TSC children. Experiments intuitively show that the new synthesis FLAIR_3_ contrast and the weighted 3D CNN strategy can effectively improve the contrast saliency of pediatric TSC lesions, and the classification performance.

The proposed deep learning method is efficient in distinguishing TSC children from healthy children and presently achieves the best performance. The proposed method has great potential in helping clinical doctors diagnose TSC children and provides an effective research tool for pediatric doctors.

## 2. Methods

### 2.1. Optimal Combination of T2W and FLAIR

Cortical and subcortical nodules are the most common lesions in TSC children. The increased prominence of lesions is crucial for clinical doctors to diagnose pediatric TSC [20]. The T2W signal is related to water content, and most of the lesions have stronger T2W signals than surrounding normal tissues, often exhibiting a bright state. Therefore, the location and size of the pediatric TSC lesions can be seen from the T2W sequence. However, the outline of the lesion is relatively vague in the T2W sequence, and it is difficult to clearly outline the outline of the lesion. Moreover, there was a strong cerebrospinal fluid (CSF) signal interference in T2W. FLAIR, also known as water-suppression imaging, suppresses (darkens) CSF hyperintensity in T2W, thereby making lesions adjacent to CSF clear (brightened). Compared with the T2W sequence, the FLAIR sequence can better represent the surroundings of the lesion and clearly show the lesion area. FLAIR is a T2W scan that selectively suppresses CSF by reversing pulses. However, CSF signal suppression comes at the expense of reducing the signal-to-noise ratio [12]. FLAIR_2_ and FLAIR_3_ have been proposed to combine T2W and FLAIR to improve lesion visualization in MS disease [12,13]. Inspired by [12,13], we propose to optimize the combination of T2W and FLAIR as a new modality named FLAIR_3_ in pediatric TSC disease as follows [13]:*FLAIR_3_* = *FLAIR^α^* × *T2W^β^**s.t. α* + *β* = 3(1)
where the optimized *α* is 1.55 and *β* is 1.45 based on the signal equations of FLAIR and T2W [13], which can optimally balance the lesion contrast between FLAIR and T2W.

### 2.2. Late Fusion Strategies

Some recent studies [21] have shown that the late fusion model could grasp the data distribution effectively and finally achieve the best classification performance. Inspired by [22,23], a weighted late fusion strategy was used to combine multi-contrast MRI for classification tasks in pediatric TSC patients. First, T2W, FLAIR, and FLAIR_3_ were fed into a feature extractor. We propose a deep weighted network (DWF net) that takes the scores of the T2W, FLAIR, and FLAIR_3_ models as input, and outputs the final classification with a simple and efficient weighted average integration method, as follows:
(2)
$$\begin{array}{c} S_{\mathrm{DWF}}\;=\;W_{1}\;\times \;S_{T2W}+W_{2}\;\times \;S_{\mathrm{FLAIR}}+W_{3}\;\times \;S_{\mathrm{FLAIR}3}\\ s.t.\;\sum_{i=1}^{3}W_{i}=1 \end{array}$$

where *S_T2W_*, *S_FLAIR_*, and *S_FLAIR3_* represent the classification scores of T2W, *FLAIR*, and *FLAIR*_3_ models, respectively. *S_DWF_* denotes the final output prediction scores of the proposed DWF-net. *W*_1_, *W*_2_, and *W*_3_ are the weights of the prediction scores of the three multi-contrast MRIs. 

To explore the optimal fusion between multi-contrast MRI and to enhance the AUC of the proposed DWF-net, the experiments were performed for values of *W*_1_ between 0 and 1, and *W*_2_ from 0.1 to 1−*W*_1_ with a step of 0.1; *W*_3_ is 1−*W*_1_−*W*_2_. The weight-searching algorithm is shown in Algorithm 1.
**Algorithm 1** The weight searching algorithm for fusion**Input:** The prediction scores *S_T2W_*, *S_FLAIR_*, and *S_FLAIR3_* of three input images and corresponding ground truth *y* on testing set.**Output:** The weight (*W_1_*, *W_2_*, and *W_3_*) with best AUC on testing set.1: Initialize *AUC _best_* ← 0.2: **for**
*i*: =0 to 10 **do**3:  **for** *j*: =0 to 10–*i*
**do**4:     *k* ← 10*-i*–*j*5:     *S _temp_ =* (*i×S_T2W_ + j×S_FLAIR_ + k×S_FLAIR3_*) *×* 0.16:     *AUC _temp_* = Compare (*S_temp_*, *y*)7:     **if**
*AUC _temp_ > AUC _best_* **then**8:      *AUC _best_* ← *AUC _temp_*9:      *W_1_* ← *i*×0.110:     *W_2_* ← *j*×0.111:     *W_3_* ← *k*×0.112:     **end for**13:  **end for**14: **end for****Return** *W_1_*, *W_2_*, and *W_3_*

### 2.3. Network Architectures

The proposed DWF-net method for pediatric TSC patients was implemented using two different 3D CNN architectures. The following sections describe two different 3D CNN models. 

ResNet was proposed in 2015 and has been widely applied in detection, segmentation, recognition, and other fields [24]. In addition, ResNet has demonstrated a stable and excellent classification performance in image classification among different variants of various 3D CNNs [24]. Therefore, the first 3D CNN model we consider is 3D-ResNet, which uses a shortcut connection to make a reference for the input of each layer and learns to form a residual function. The residual function is easier to optimize, making the number of network layers much deeper, and can easily obtain a higher accuracy from deeper depths.

For the second 3D CNN model, we utilized the 3D-EfficientNet architecture [25] as our feature extractor. This classification network is known for its efficiency in improving accuracy and reducing the training time and network parameters. The EfficientNet was designed using a neural architecture search and employs the mobile inverted bottleneck convolution (MBConv) module as its core structure. This module, similar to depth-wise separable convolution, minimizes parameters significantly. In addition, the attention idea of the squeeze-and-excitation network (SENet) is also introduced [26] in EfficientNet. The attention mechanism of SENet allows the model to focus more on channel features that are most informative, while suppressing those unimportant channel features, thereby improving the model performance.

As shown in Figure 1a, for the pediatric TSC identification tasks with one single MRI modality, the 3D-ResNet34 and 3D-EfficientNet were used as a feature extractor. When DWF-net was used, two or three modalities were applied as inputs, as shown in Figure 1b. Table 1 displays the 10 models that were trained in this study, each with distinct architectures and inputs.

## 3. Materials and Experiments

### 3.1. Dataset

In this study, all pediatric volunteers were from Shenzhen Children’s Hospital. The study was approved by the Ethics Committee of Shenzhen Children’s Hospital (No.2019005). Written informed consent was obtained from all pediatric volunteers and/or their parents. In total, 349 TSC children and 331 healthy children (HC) were included in this study. Inclusion criteria for pediatric TSC patients were (1) aged 0–20 years, (2) no other neurological disorders, and (3) clinically diagnosed with TSC. (4) T2W and FLAIR images are complete and clear. Inclusion criteria for healthy children were (1) aged 0–20 years, (2) without any neurological disorder, (3) clinically defined normal or non-specific findings during routine clinical care. (4) T2W and FLAIR images are complete and clear. Figure 2 shows the exclusion and inclusion criteria of our study.

The data were randomly split into train-validation-test sets in a 7:1:2 ratio. To ensure that every group had the same class proportion, stratified random sampling was employed. Training, validation, and testing datasets had no overlap of patients.

### 3.2. Data Processing

Firstly, a FMRIB Linear Image Registration Tool (FLIRT) of FSL (http://fsl.fmrib.ox.ac.uk (accessed on 1 January 2021.)) was used to register T2W into the FLAIR space, and mutual information was used as the cost function. In neuroimaging studies, the lesions are usually located in the brain tissue, and the skull part is an irrelevant site. When brain MRI images are used for classification network research, the brain tissue of the region of interest is often the input. HD-bet is an algorithm for extracting brain tissue [27], which can remove irrelevant images such as of the neck and eyeball. Therefore, in the second step, the deep learning tool HD-bet is used to strip the skull in MRI. Subsequently, all 3D MRI images were resized to 128 × 128 × 128, and the image intensity was normalized to the range of 0 to 1 using the min–max normalization formula:
(3)
$$x_{Normalized}=\frac{x-Min(x)}{Max(x)–Min(x)}$$

where *Max*(*x*) and *Min*(*x*) represent the highest and lowest values of the brain-extracted MRI images, respectively, and *x_Normalized_* refers to the normalized MRI images. Finally, T2W and FLAIR were combined and transformed into FLAIR_3_. The flowchart illustrating the data preprocessing can be found in Figure 3.

### 3.3. Baseline and Effectiveness of Skull Stripping

In this study, we compared 10 different proposed 3D CNN models with a 2D-InceptionV3 model [18] (baseline model) to evaluate the effectiveness of the proposed deep learning methods. The 2D-InceptionV3 model was exclusively trained on our FLAIR data, with the maximum transverse slice of the FLAIR chosen as the input. Furthermore, we conducted a series of experiments on FLAIR images and T2W images with and without skull-stripping preprocessing to assess the effectiveness of the skull-stripping methodology.

### 3.4. Comparison of Normalization Methods

Typically, normalization methods often have a significant impact on the performance of deep learning models. The min–max normalization and Z-score normalization are most used in medical image normalization. While the min–max normalization approach is appropriate for most kinds of data and can effortlessly maintain the initial data distribution structure, it is not ideal for handling sparse data and is prone to being affected by outliers. The Z-score normalization method employs the mean and standard deviation of the original data to normalize it. The following formula illustrates this:
(4)
$$x_{Normalized}=\frac{x-Mean(x)}{std(x)}$$


When 
$Mean\left(x\right)$
= 0, 
std(x)
 = 1, that is, the mean is 0 and the standard deviation is 1, meaning that the processed data conform to the standard normal distribution. This Z-score method is suitable for most types of data, but it is a centralized method, which will change the distribution structure of the original data, and it is also not suitable for the processing of sparse data. To explore the effectiveness of the normalization operation, we conducted three sets of experiments on both T2W and FLAIR images when using the same network, which are without the normalization method, the Z-score normalization, and the min–max normalization, respectively.

### 3.5. Model Training and Evaluation

For our experiments, we used the same partitioning for the training set, validation set, and test set across all models. Each model was trained using a learning rate of 0.0001, SGD optimization, a batch size of 4, and 50 epochs, with the binary cross-entropy loss function. To implement the training, validation, and testing process, we used Python version 3.8.10 and PyTorch version 1.9.0 environments.

For each cohort, we calculated the area under the curve (AUC) of the receiver operating characteristic (ROC), accuracy (ACC), sensitivity (SEN), and specificity (SPE) to evaluate the classification performance of all models. These metrics rely on the true positive (TP), which counts the total number of correct positive classifications, and the true negative (TN), which represents the total number of accurate negative classifications. The false positive (FP) accounts for the total number of positive classifications that are incorrect, while the false negative (FN) represents the total number of negative classifications that are incorrect. We obtained the ACC, SEN, and SPE through the following formulas:

Accuracy (ACC): The percentage of the whole sample that is correctly classified: 
(5)
$$ACC=\frac{TP+TN}{TP+TN+FP+FN}$$


Sensitivity (SEN): The percentage of the total sample that is true and correctly classified: 
(6)
$$SEN=\frac{TP}{TP+FN}$$


Specificity (SPE): The percentage of the total sample that is negative and correctly classified: 
(7)
$$SPE=\frac{TN}{TN+FP}$$


### 3.6. Statistical Analysis

For this research, categorical variables were presented using the frequency and percentage, while continuous variables were expressed as the mean ± standard deviation. Continuous variables were analyzed using the F-test, while categorical variables underwent a chi-square analysis. Statistical significance was defined as *p* < 0.05. All statistical analyses were performed using the scikit learn, scipy, and stats libraries in Python 3.8.10.

## 4. Results

### 4.1. Clinical Characteristics of Patients

All of the 680 child subjects’ primary clinical features are listed in Table 2. Among the 349 TSC patients, 188 (53.9%) were identified as male, averaging 45.5 months in age. Moreover, among the 331 HC, 183 (55.3%) were identified as male, averaging 733 months in age. There was a significant difference in the average age between the HC group and the TSC group, with a *p*-value less than 0.05. There was no significant difference in gender.

### 4.2. Visualization Results of FLAIR_3_

Figure 4 shows FLAIR, T2W, and FLAIR_3_ images of a TSC child and a healthy child. On three MRI images of the TSC child, it can be observed that the contrast between the lesions and brain tissue on FLAIR is not clear enough, there is a severe interference of cerebrospinal fluid on T2W, and the contrast and clarity of the lesions on the newly generated FLAIR_3_ image are significantly improved (TSC lesion as shown by the red arrow). In addition, FLAIR_3_ inhibits cerebrospinal fluid and can clearly locate the TSC lesion.

### 4.3. Performance of the Models

The performance of DWF-net varies with the weight of W_1_, W_2_, and W_3_ as shown in Figure 5. The feature extractor in Figure 5a is 3D-EfficientNet, and the best AUC performance of 3D-EfficientNet is 0.989 (W_1_ = 0.0, W_2_ = 0.3, W_3_ = 0.7). Among the models evaluated, Res_DWF_net (with weight parameters W_1_ = 0.2, W_2_ = 0.3, W_3_ = 0.5), which employs 3D-ResNet as a feature extractor and a late fusion strategy as depicted in Figure 5b, achieves the highest performance. This model has an accuracy of 0.985 and an AUC of 0.998, outperforming other models.

The results for all the compared models in the testing dataset are presented in Table 3. When using 3D-EfficientNet, FLAIR_3_ achieves an AUC performance of 0.987 and the AUC of Eff_FLAIR_T2W is 0.974, and the AUC of FLAIR_3_ is higher than Eff_FLAIR_T2W. FLAIR_3_ achieves an AUC performance of 0.997 when using 3D-ResNet as the feature extraction network. When the feature extraction network is 3D ResNet, the AUC of Res_FLAIR_T2W is 0.994, and the AUC of FLAIR_3_ is higher than Res_FLAIR_T2W.

When using the same single-modal MRI as inputs, 3D-ResNet outperforms 3D-EfficientNet. Additionally, the AUC performance of the FLAIR_3_ model outperforms the T2W-only model and FLAIR-only model. The baseline network (InceptionV3) achieves an AUC performance of 0.952, and the performance of our all-3D network exceeds the AUC performance of the baseline network of InceptionV3.

ROC curves for all models of the testing cohort are shown in Figure 6a–c, and Figure 6d shows the classification performance for all models of the testing cohort.

### 4.4. Results of Skull Stripping

The classification performance of FLAIR and T2W images, with or without skull dissection, is presented in Table 4. The table demonstrates that if the network structure and input modality remain constant and the skull dissection preprocessing is not carried out, the classification performance of 3D ResNet and 3D EfficientNet will show a decline.

### 4.5. Comparison of Normalization Methods

Table 5 and Figure 7 depict the classification performance of three normalization methods, including without normalization, Z-score normalization, and min–max normalization on FLAIR images and T2W images. The horizontal axis represents the different normalization techniques, while the vertical axis represents their corresponding performance. In instances where the input modality and network structure remain constant, it is worth noting that the without-normalization method has the poorest AUC performance. Furthermore, the AUC performance of the min–max normalization technique is better than the Z-score normalization technique.

## 5. Discussion

The main objective of the proposed approach is to identify TSC children at an early stage using a 3D CNN model in conjunction with multi-contrast MRI in an automated manner. Initially, the approach incorporates FLAIR_3_ as a novel modality for diagnosing pediatric TSC lesions and optimizes the T2W and FLAIR combination to enhance the lesion–brain contrast in a clinic. The findings indicate that FLAIR_3_ has the ability to enhance the prominence of TSC lesions, while also enhancing classification accuracy and providing a more intuitive understanding of our deep learning model. Otherwise, the proposed method used two networks as feature extractors; one is 3D-EfficientNet, which is a parameter-efficient deep convolutional neural network framework, and the other classification network is 3D-ResNet, which is a classical residual network. Previously, the FLAIR_3_ modality was only used in MS disease [13], but the proposed methods generalized it to pediatric TSC disease and demonstrated that FLAIR_3_ was able to better visualize TSC lesions. Furthermore, a multi-modal fusion network for multi-contrast MRI data was proposed, which can feed FLAIR_3_ as a new modality into the proposed DWF-net network, finally achieving a state-of-the-art classification performance in identifying children with pediatric TSC. And the dataset has no PET and EEG as input, and only has just the structural MRI that can be easily and wildly collected at any hospital, which helpfully maximizes the potential applicability of the proposed approach in clinical practice. In summary, the proposed method also has innovations in the following aspects: 1) the use of a weighted fusion algorithm to maximize the fusion multi-contrast MRI and optimize weights to improve performance; 2) firstly proposes to use a FLAIR_3_ image to position and visualize the lesions in a clinical diagnosis of TSC. 3) The utilization of FLAIR_3_ as the complementary imaging input to maximize the information extracted from the structure MRI.

In comparison to the 2D CNN model InceptionV3 discussed in [18], the proposed 3D CNN models exhibit an enhanced classification performance. Some previous studies are also consistent with our conclusion that 3D networks perform better than 2D networks [19,28]. We believe that the performance improvement of the 3D network is mainly due to the full use of the spatial features of MRI voxels, which can extract more information. In this study, the proposed late fusion method can improve the classification performance compared to a single modality using a 3D CNN approach, implying that combining multiple contrasting MRI can exploit complementary visual information between multiple sequences. This result is consistent with a recent study by Han Peng et al. [29], which demonstrated that combining models from diverse modalities with complementary information leads to a superior performance. The success of the ensemble strategy is not only attributed to the number of large models but also to independent information gathered from different modalities. Additionally, recent research has revealed that the late fusion method outperforms the early fusion technique [30,31]. In addition, Jonsson et al. used a majority voting strategy to form the final predictions and achieved performance gains with multimodal inputs [22]. In our experimental results, our findings indicate that when utilizing the same MRI modality as network inputs, all models with 3D-ResNet feature extractors outperform the 3D-EfficientNet model. One possible explanation is that 3D-ResNet has more network parameters than 3D-Effectient, and the network structure is more complicated. Therefore, 3D-ResNet can extract more high-level image feature information than 3D-EfficientNet.

Surprisingly, our experiments have successfully demonstrated the effectiveness of FLAIR_3_ in a pediatric TSC diagnosis, and the AUC performance of the FLAIR_3_-only model outperforms the T2W-only model and FLAIR-only model when using the same network. We found that the use of 3D-EfficientNet results in a better AUC score for the Eff_FLAIR_3_ model compared to the Eff_FLAIR_T2W model and that the Res_FLAIR_3_ model outperforms the Res_FLAIR_T2W model when using the feature extraction network 3D ResNet. This could imply that FLAIR_3_ can provide more information. When the late fusion strategy is used, the weight W_3_ of FLAIR_3_ is the largest. A reasonable note is that FLAIR_3_ can enhance the lesion-to-brain contrast and the TSC lesion is clearer in FLAIR_3_ than in T2W and FLAIR, so FLAIR_3_ can offer more low-dimensional visual lesion information for deep learning during the feature extraction stage. Such low-dimensional visual information may be very helpful for our deep learning algorithms, which could increase the interpretability of our deep learning algorithms [32].

Moreover, skull stripping plays a crucial role in computational neuro-imaging by being a vital preprocessing step that has a direct impact on subsequent analyses [33,34,35]. In this study, we found that both the 3D-ResNet and 3D-EfficientNet models perform better when utilizing MRI with skull stripping applied as the input. This may be due to the fact that the pixel value of the skull is significantly higher than that of the brain tissue [30,36], which allows for more information to be extracted during the feature selection phase. However, it is important to note that such information may be irrelevant for our deep learning methods and may even reduce their performance [37].

Furthermore, image normalization is critical to develop powerful deep learning methods [38,39]. In this study, the experiments included normalization, no normalization, min–max normalization, and Z-score normalization. All of the results showed that the AUC performance without the normalization method is the worst; the AUC performance of the min–max normalization is better than the Z-score normalization when the input modality and network structure are the same. Therefore, we suggest that in future similar studies, the min–max normalization method can be used as a primary choice to normalize the MRI images.

Otherwise, many experts considered that tubers are stable in size and appearance after birth and that the proportion to the whole brain will not obviously change with age [40]. The myelination process in a clinic has three stages, namely before 7–8 months of age, 7–8 months to 2 years of age, and after 2 years of age. So, the TSC situation of MRI after 2 years of age should be the same as before, but myelination after 2 years of age may not have affected our MRI images [41]. But these are statistical results, and there are some different situations for different TSC patients. In a clinic, MRI should be scanned several times under the age of 2 to reflect dynamic changes in epileptic lesions. Here, we did not exclude children under 2 years of age for being close to real clinical situations. The deep learning method we proposed can be promoted in a clinic and only needs to collect FLAIR and T2W images of a patient. Our method is simple and effective in a clinic and can be used as a computer-aided tool to help doctors diagnose TSC patients. In the future, further situations of TSC patients should be evaluated.

## 6. Conclusions

In summary, a novel deep learning method of the weighted late fusion model was proposed to effectively diagnose pediatric TSC lesions with multi-contrast and synthesis-contrast FLAIR_3_ MRI. The collected dataset of pediatric TSC disease has a total of 680 children, including 331 healthy and 349 TSC children. The current testing results illustrated that the proposed approach can attain a state-of-the-art AUC of 0.998 and accuracy of 0.985. As such, this method can act as a robust foundation for future studies regarding pediatric TSC patients.

## 7. Patents

The work reported in this manuscript has resulted in a patent.

## Figures and Tables

**Figure 1 bioengineering-10-00870-f001:**
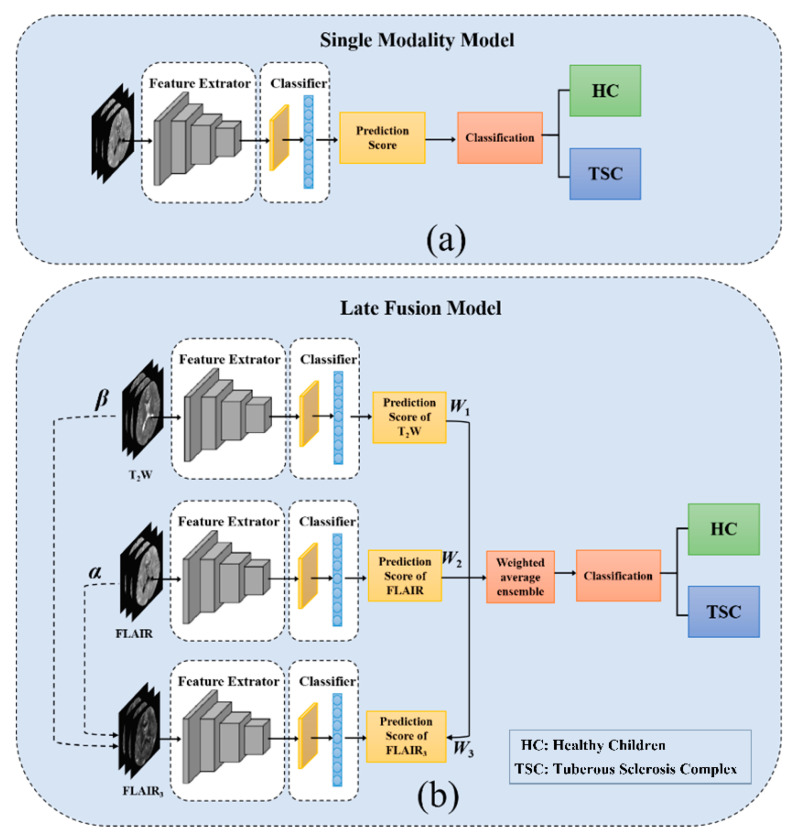
Overall network structure, (**a**) single modality model pipeline, (**b**) schematic of the proposed DWF-net pipeline. The two dotted lines represent the optimal combination of T2W and FLAIR to generate FLAIR_3_.

**Figure 2 bioengineering-10-00870-f002:**
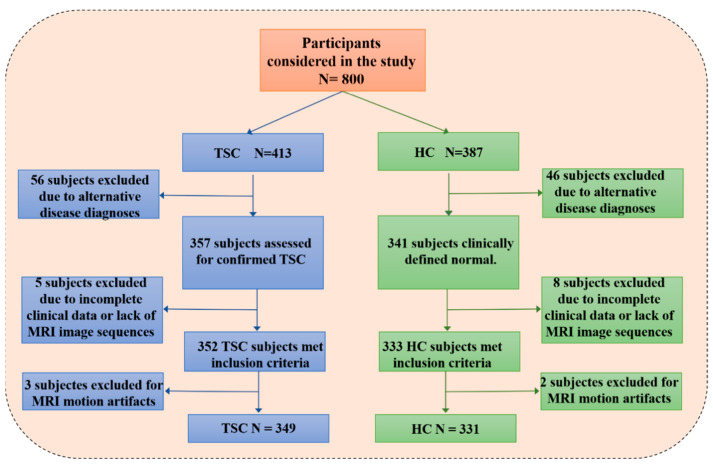
Study exclusion and inclusion criteria of the pediatric dataset.

**Figure 3 bioengineering-10-00870-f003:**
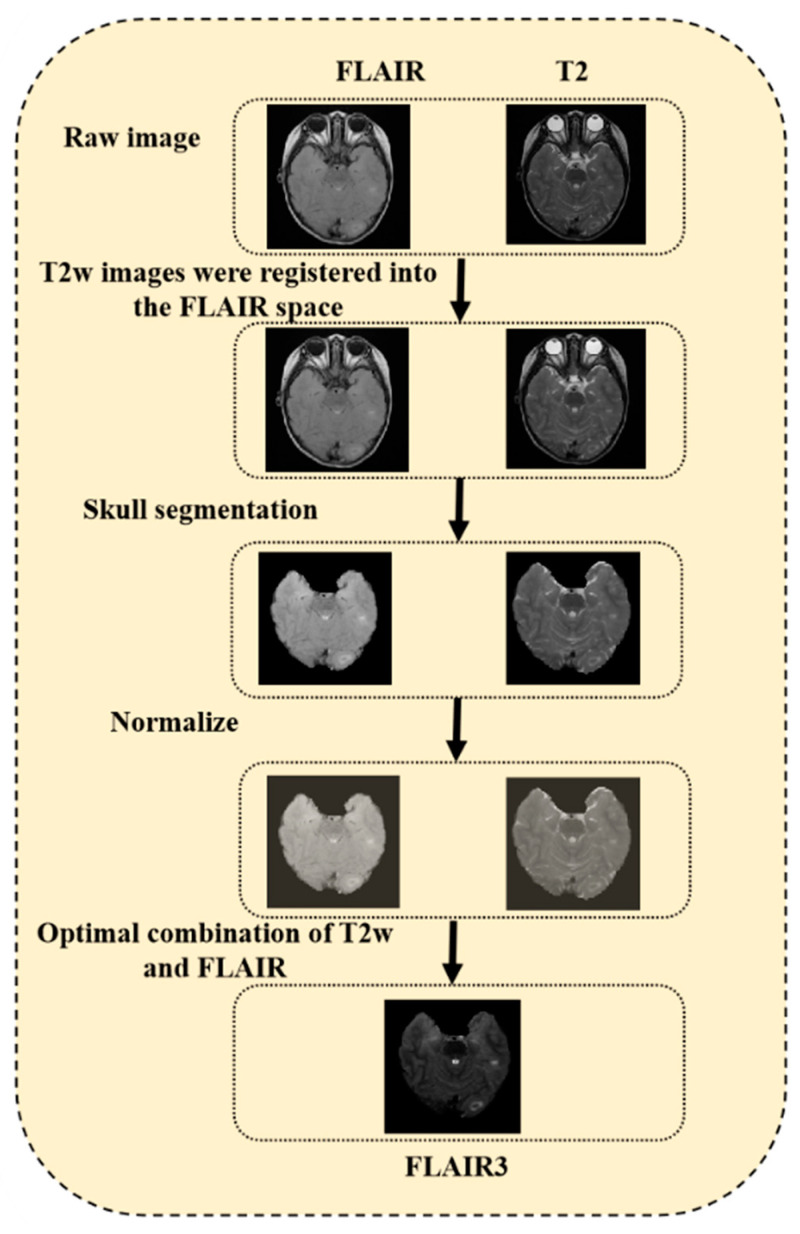
Flowchart of the data preprocessing.

**Figure 4 bioengineering-10-00870-f004:**
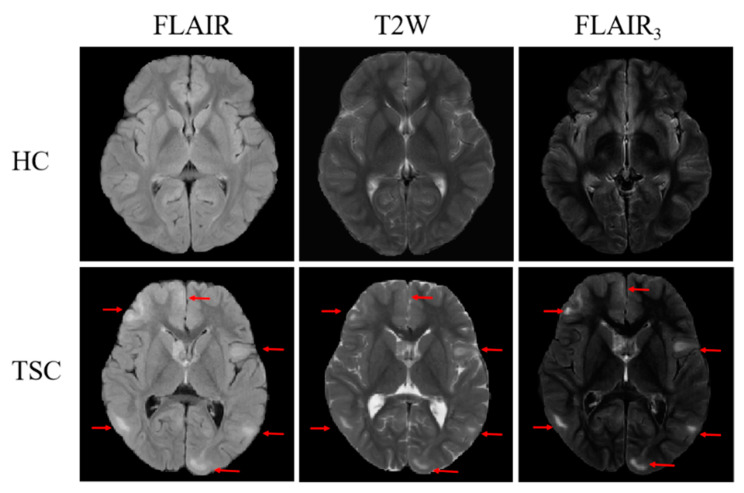
Representative MRI from a TSC child and a healthy child, including T2W, FLAIR, and the proposed FLAIR_3_ (the red arrow highlights the TSC lesion).

**Figure 5 bioengineering-10-00870-f005:**
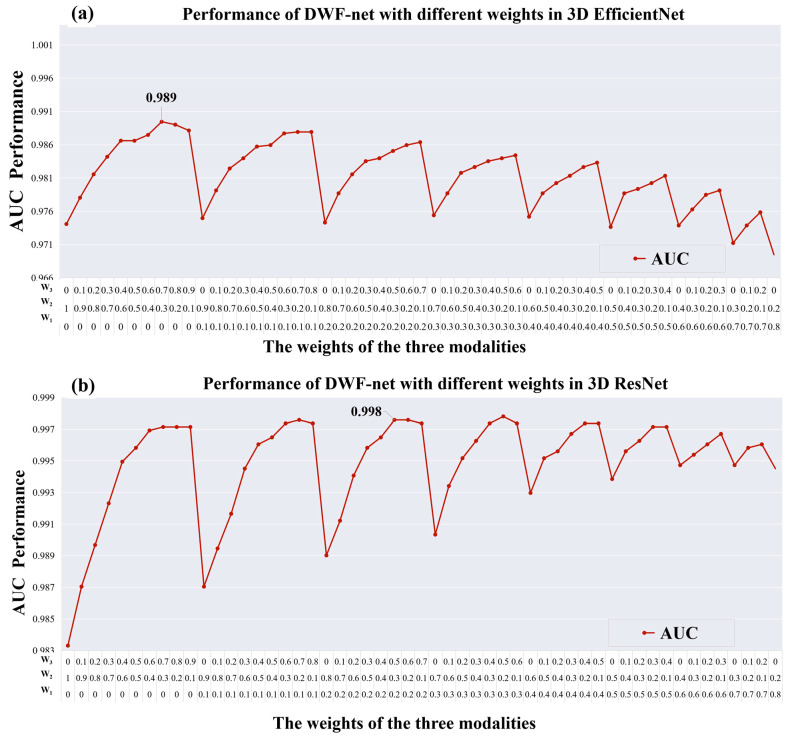
The performance of DWF-net with different weights. The feature extractor in (**a**) is 3D-EfficientNet, and the feature extractor in (**b**) is 3D-ResNet. The horizontal axis represents the weight of W_1_, W_2_, and W_3_, and the vertical axis represents the performance of AUC.

**Figure 6 bioengineering-10-00870-f006:**
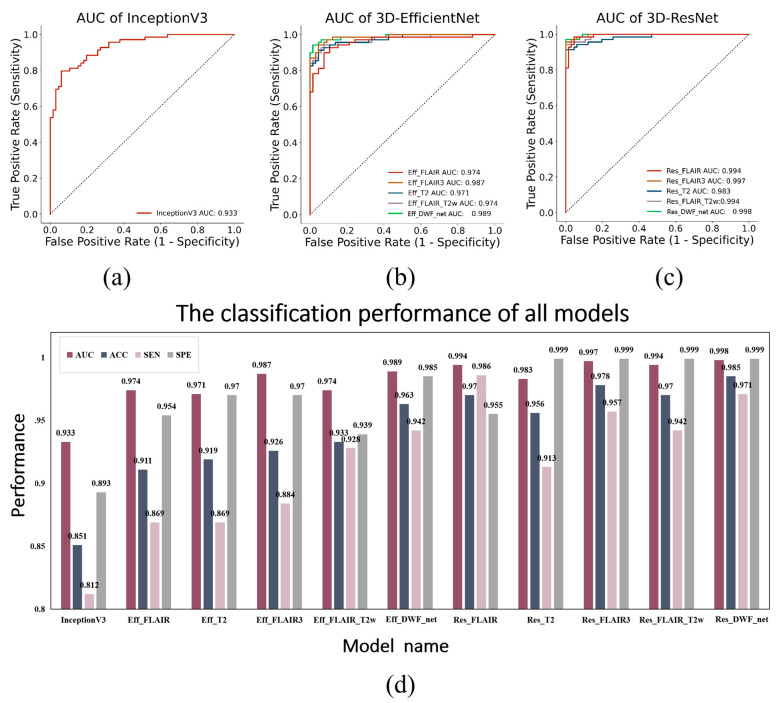
(**a**–**c**) represent the ROC curves for all models of the testing cohort. (**d**) represents the classification performance for all models of the testing cohort. The horizontal axis shows the model name, while the vertical axis represents the performance regarding AUC, ACC, SEN, and SPE.

**Figure 7 bioengineering-10-00870-f007:**
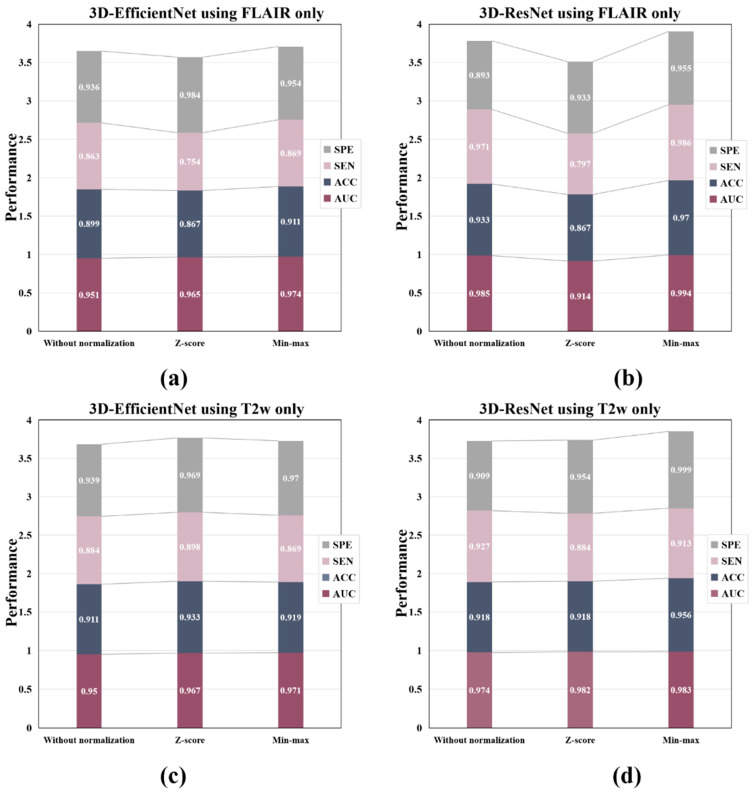
The classification performance of the without-normalization method, the Z-score normalization, and the min–max normalization in FLAIR images and T2W images. (**a**) 3D-EfficientNet as a network feature extractor, FLAIR as the network input. (**b**) 3D-ResNet as a network feature extractor, FLAIR as the network input. (**c**) 3D-EfficientNet as a network feature extractor, T2W as the network input. (**d**) 3D-ResNet as a network feature extractor, T2W as the network input.

**Table 1 bioengineering-10-00870-t001:** Detailed information on ten network structures.

Model Name	Input Modality	Method
Eff_FLAIR	FLAIR only	3D-EfficientNet
Eff_T2W	T2W only	3D-EfficientNet
Eff_FLAIR_3_	FLAIR_3_ only	3D-EfficientNet
Eff_FLAIR_T2W	FLAIR + T2W	DWF_net
Eff_DWF_net	FLAIR + T2W + FLAIR_3_	DWF_net
Res_FLAIR	FLAIR only	3D-ResNet34
Res_T2	T2W only	3D-ResNet34
Res_FLAIR_3_	FLAIR_3_ only	3D-ResNet34
Res_FLAIR_T2W	FLAIR + T2W	DWF_net
Res_DWF_net	FLAIR + T2W + FLAIR_3_	DWF_net

**Table 2 bioengineering-10-00870-t002:** The main clinical characteristics of all 680 child subjects.

	TSC	HC	*p*-Value
Number	349	331	-
Male, number (%)	188 (53.9%)	183 (55.3%)	0.711
Age at imaging, mean ± SD (months)	45.5 ± 46.6	73.3 ± 49.2	<0.001

**Table 3 bioengineering-10-00870-t003:** Detailed performance of different models in pediatric testing datasets.

Input Modality	Model Name	AUC	ACC	SEN	SPE
FLAIR + T2W	InceptionV3 [18]	0.933	0.851	0.812	0.893
FLAIR only	Eff_FLAIR	0.974	0.911	0.869	0.954
T2W only	Eff_T2W	0.971	0.919	0.869	0.970
FLAIR_3_	Eff_FLAIR_3_	0.987	0.926	0.884	0.970
FLAIR + T2W	Eff_FLAIR_T2W	0.974	0.933	0.928	0.939
FLAIR + T2W + FLAIR_3_ (*W_1_ =* 0.0, *W_2_ =* 0.3, *W_3_ =* 0.7)	Eff_DWF_net	**0.989**	**0.963**	**0.942**	**0.985**
FLAIR only	Res_FLAIR	0.994	0.970	**0.986**	0.955
T2W only	Res_T2W	0.983	0.956	0.913	0.999
FLAIR_3_	Res_FLAIR_3_	0.997	0.978	0.957	0.999
FLAIR + T2W	Res_FLAIR_T2W	0.994	0.970	0.942	0.999
FLAIR + T2W + FLAIR_3_ (*W_1_ =* 0.2, *W_2_ =* 0.3, *W_3_ =* 0.5)	Res_DWF_net	**0.998**	**0.985**	0.971	**0.999**

**Table 4 bioengineering-10-00870-t004:** The results of with/without skull stripping in T2W and FLAIR.

Modality	Model Name	Preprocessing	AUC	ACC	SEN	SPE
FLAIR only	3D-EfficientNet	Without skull stripping	0.898	0.829	0.754	0.909
		Skull stripping	**0.974**	**0.911**	**0.869**	**0.954**
	3D-ResNet	Without skull stripping	0.959	0.881	0.855	0.909
		Skull stripping	**0.994**	**0.970**	**0.986**	**0.955**
T2W only	3D-EfficientNet	Without skull stripping	0.968	0.916	**0.881**	0.951
		Skull stripping	**0.971**	**0.919**	0.869	**0.970**
	3D-ResNet	Without skull stripping	0.914	0.829	0.797	0.863
		Skull stripping	**0.983**	**0.956**	**0.913**	**0.999**

**Table 5 bioengineering-10-00870-t005:** The classification performance of with/without skull stripping in FLAIR images and T2W images.

Modality	Model Name	Preprocessing	AUC	ACC	SEN	SPE
FLAIR only	3D-EfficientNet	Without normalization	0.951	0.899	0.863	0.936
		Z-score	0.965	0.867	0.754	**0.984**
		Min–max	**0.974**	**0.911**	**0.869**	0.954
	3D-ResNet	Without normalization	0.985	0.933	0.971	0.893
		Z-score	0.914	0.867	0.797	0.933
		Min–max	**0.994**	**0.970**	**0.986**	**0.955**
T2W only	3D-EfficientNet	Without normalization	0.950	0.911	0.884	0.939
		Z-score	0.967	**0.933**	**0.898**	0.969
		Min–max	**0.971**	0.919	0.869	**0.970**
	3D-ResNet	Without normalization	0.974	0.918	**0.927**	0.909
		Z-score	0.982	0.918	0.884	0.954
		Min–max	**0.983**	**0.956**	0.913	**0.999**

## Data Availability

All of our data is from Shenzhen Children’s Hospital and the data are unavailable due to privacy or ethical restrictions.

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
