# Peer review of "Recognizing Pediatric Tuberous Sclerosis Complex Based on Multi-Contrast MRI and Deep Weighted Fusion Network"

_bioengineering, 2023, doi:10.3390/bioengineering10070870_

Round 1

Reviewer 1 Report

The manuscript presents a tuberous method based on deep neural networks for the early diagnosis of tuberous sclerosis complex in children from multi-contrast MRI scans. The manuscript is well-written, methodologically sound, and addresses an interesting topic. However, some revisions are needed to improve the quality of work and clarify its potential impact in the clinical field.

Comments for improvement are reported in the following.

Methods:

- Paragraph 3.6: the choice for the statistical tests should be clarified. What is the purpose of the statistical analysis? What are the considered variables? What is the rationale for the tests that have been carried out? 

- Line 249: "P-value" should be written as "p-value".

Results:

- Lines 259-260: please verify the average age mentioned for the HC group. it does not match the values reported in Table 2.

- Table 2 could be commented on with more detail. How did the authors address the significant difference in the mean age? Can age be a confounding factor in the analysis?

- Figure 6 is redundant as it reports the same values displayed in Table 3. The same occurs with Figure 7 and Table 5.

Discussion:

- While a proper comparison with the literature is carried out and discussed, no interpretation and explanation of the deep learning model are provided. As the explainability of black box AI models is a fundamental step for clinical translation, I would suggest considering also discussing potential explainability routes to assess the proposed model.

- The authors should better discuss the potential clinical impact and translation. What would be the requirements for implementing the proposed methodology in the clinical field? What could be the pros and cons of the method?

Reviewer 2 Report

In this study, Jiang et al proposed an AI system to recognize brain lesions of tuberous sclerosis complex (TSC) using multi-contrast MRI (a new imaging modality named FLAIR3) and deep learning (the weighted late fusion model), based on the data of 349 TSC children and 331 healthy children. According to the authors, this study is the first to apply the combination of deep learning and multi-contrast MRI on the diagnosis of MRI.

From the viewpoint of a pediatric neurologist, the reviewer sees in this manuscript some issues:

1.    There are in the title and text very many phrases of “rare TSC”, which sounds redundant and even annoying. TSC is indeed rare, but not ultra-rare, with prevalence of approximately 1/6,000-10,000. The authors recruited as many as 413 TSC patients in a single hospital. Furthermore, the rarity of the condition has nothing to do with the significance of this study.

2.    In the first paragraph of Introduction, the authors emphasize the impact of TSC epileptic seizures in the first year of life. However, the average year of TSC patients in this study is as high as 45.5 months. It is also unclear whether the methodology of this study is as useful in infants under 2 years of age whose brains are undergoing myelination as children older than 2 years.

3.    In the second paragraph of Introduction, the authors point out that a strong signal of CSF in T2-weighted imaging interferes the visualization of periventricular TSC lesions. However, the most important information provided by cranial MRI is subcortical high T2 signals, as indicated in Figure 4 (red arrows). 

4.    The abbreviation “HC” should be spelt out not only in the text (Section 3.1) but also in the legend to Figure 1.

5.    There is a syntax error: Some recent studies [have] shown (line 112).

In general, this article is well written.

Round 2

Reviewer 1 Report

The authors have addressed all previous concerns.

Please revise this:

- "...Statistical significance was defined as P<0.05...", please use "p" instead of "P" for indicating the p-value

Reviewer 2 Report

In this revised version, authors have duly addressed most of the issues in the original version.

Generally well written.